# Advanced computational approaches for predicting sunflower yield: Insights from ANN, ANFIS, and GEP in normal and salinity stress environments

**Sanaz Khalifani[1], Reza Darvishzadeh[1]\*, Majid Montaseri[2], Sarvin Zaman Zad Ghavidel[2], Hamid Hatami Maleki[3], Mojtaba Kordrostami**[ID][4]\*

**1** Department of Plant Production and Genetics, Faculty of Agriculture, Urmia University, Urmia, Iran, **2** Department of Water Engineering, Faculty of Agriculture, Urmia University, Urmia, Iran, **3** Department of Plant Production and Genetics, Faculty of Agriculture, University of Maragheh, Maragheh, Iran, **4** Nuclear Agriculture Research School, Nuclear Science and Technology Research Institute (NSTRI), Karaj, Iran

\* r.darvishzadeh@urmia.ac.ir (RD); mkordrostami@aeoi.org.ir (MK)

## Abstract

Prediction of crop yield is essential for decision-makers to ensure food security and provides valuable information to farmers about factors affecting high yields. This research aimed to predict sunflower grain yield under normal and salinity stress conditions using three modeling techniques: artificial neural networks (ANN), adaptive neuro-fuzzy inference system (ANFIS), and gene expression programming (GEP). A pot experiment was conducted with 96 inbred sunflower lines (generation six) derived from crossing two parent lines, over a single growing season. Ten morphological traits—including hundred-seed weight (HSW), number of leaves, leaf length (LL) and width, petiole length, stem diameter, plant height, head dry weight (HDW), days to flowering, and head diameter—were measured as input variables to predict grain yield. Salinity stress was induced by applying irrigation water with electrical conductivity (EC) levels of 2 dS/m (control) and 8 dS/m (stress condition) using NaCl, applied after the seedlings reached the 8-leaf stage. The GEP model demonstrated the highest precision in predicting sunflower grain yield, with coefficient of determination ($R^2$) values of 0.803 and 0.743, root mean squared error (RMSE) of 4.115 and 4.022, and mean absolute error (MAE) of 3.177 and 2.803 under normal conditions and salinity stress, respectively, during the testing phase. Sensitivity analysis using the GEP model identified LL, head diameter, HSW, and HDW as the most significant parameters influencing grain yield under salinity stress. Therefore, the GEP model provides a promising tool for predicting sunflower grain yield, potentially aiding in yield improvement programs under varying environmental conditions.

## 1. Introduction

Sunflower (*Helianthus annuus* L.), belonging to the Compositae family and native to North America, is a globally significant oilseed crop [1]. With a cultivated area of 27.8 million hectares and a total production of 50.3 million tons worldwide, sunflowers account for

**Data availability statement:** All relevant data necessary to replicate the findings of this study are provided within the paper. This includes the metadata and detailed methods. The dataset used for training and testing the models (216 and 72 samples, respectively) under both normal and salinity stress conditions, as well as supplementary tables and figures that provide additional context and results, are included. For further clarifications or specific data requests, please contact the vice president for research and technology at Urmia University, email: info@urmia.ac.ir. This contact person is responsible for facilitating data access and addressing inquiries related to the data used in this study.

**Funding:** The author(s) received no specific funding for this work.

**Competing interests:** The authors have declared that no competing interests exist.

approximately 8% of the global oilseed trade [2]. Sunflower seeds contain 35–42% oil and 20% protein, and are rich in phenolic compounds, flavonoids, polyunsaturated fatty acids, and vitamins, endowing them with antioxidant, antimicrobial, anti-inflammatory, antihypertensive, and wound-healing properties [3].

Despite its economic importance, sunflower production is adversely affected by both biotic and abiotic stresses throughout its phenological stages. Salinity stress, in particular, poses a significant challenge, as sunflower cultivars exhibit a wide range of tolerance levels—from highly sensitive to semi-tolerant [4,5]. Salt stress can lead to a reduction in hypocotyl length and root proliferation, browning of root tips, loss of cotyledon development, leaf necrosis in young plants, impaired inflorescence development, and difficulties in seed formation [4]. Moreover, salinity induces the production of reactive oxygen species, including singlet oxygen, superoxide radicals, hydroperoxy radicals, hydrogen peroxide, and hydroxyl radicals, resulting in secondary oxidative stress [6]. The detrimental effects of salinity on plant growth are associated with decreased osmotic potential of the soil solution, ionic imbalances, specific ion toxicity, or a combination of these factors, leading to complex physiological, biochemical, and molecular disruptions [7].

Several studies have explored the impact of salinity stress on sunflower genotypes. Ebeed, Hassan [8] evaluated sunflower genotypes under varying degrees of soil salinity in Egypt and found that high salinity levels significantly reduced grain yield, oil content, and altered biochemical attributes across all genotypes. Anwar-ul-Haq, Akram [9] studied the morpho-physiological and biochemical characteristics of three sunflower genotypes under different salinity levels and reported substantial effects on plant height, biomass, leaf area, and ionic concentrations. Ghaffari, Fanaei [10] assessed 24 inbred lines for salt tolerance and observed significant reductions in seed and oil yields by 34% and 31%, respectively. Gogna, Choudhary [11] investigated lipid composition changes under salt stress and noted that salt-sensitive varieties experienced a considerable decrease in the unsaturated/saturated fatty acids ratio due to $Na^+$ accumulation.

Predicting crop yield under stress conditions is crucial for agricultural planning and ensuring food security. Traditional approaches, such as multiple linear regression (MLR), have been widely used to model agricultural processes [12]. However, MLR models often face limitations due to significant nonlinear and multicollinear relationships among variables, as well as genotype-environment interactions, rendering them inefficient for yield prediction [13–15].

To overcome these limitations, artificial intelligence (AI) techniques, particularly artificial neural networks (ANNs), have been employed for yield modeling. ANNs are capable of capturing complex nonlinear relationships between input variables and yield [16–19]. For instance, Piekutowska et al. [16] successfully predicted early potato yield using ANNs, while Rajković et al. [17] applied ANNs to estimate yield and quality traits in winter rapeseed. Similarly, Niazian et al. [18] reported superior performance of ANNs over MLR in modeling seed yield of *Trachyspermum ammi* L.

Despite the successes, ANNs possess inherent limitations, such as the risk of overfitting, lack of interpretability due to their "black-box" nature, and challenges in capturing complex nonlinear interactions among variables [19]. These limitations can hinder the practical application of ANN models in agricultural decision-making, where understanding the relationships between variables is essential.

To address these gaps, alternative modeling techniques like adaptive neuro-fuzzy inference systems (ANFIS) and gene expression programming (GEP) have been proposed. ANFIS combines the learning capabilities of neural networks with the reasoning abilities of fuzzy logic, effectively handling uncertainty and complex nonlinear systems [20]. It provides interpretable models through fuzzy rules, enhancing the understanding of input-output relationships. GEP,

on the other hand, evolves explicit mathematical expressions, offering clear insights into the underlying relationships among variables and improving model interpretability [21]. Both methods have shown promise in various fields but have not been extensively applied to predict sunflower grain yield under salinity stress.

Therefore, this study aims to model sunflower grain yield under normal and salinity stress conditions using ANN, ANFIS, and GEP models. By comparing the efficiency of these models in predicting seed yield and identifying the most influential parameters affecting grain yield, we seek to provide more accurate and interpretable tools for yield prediction. This could significantly aid breeders and farmers in making informed decisions to enhance sunflower production under challenging environmental conditions.

## 2. Materials and methods

### 2.1. Field experiments and data collection

This study investigated the effect of salinity stress on 96 inbred sunflower lines (generation six) derived from crossing two lines: PAC2 (♀) and RHA266 (♂). The French National Institute of Agronomic Research (INRA) prepared the recombinant inbred lines (RILs) using the single-seed descent selection method. The paternal line RHA266 is a cross between wild genotypes *Helianthus annuus* and Peredovik, developed by the United States Department of Agriculture, while the maternal line PAC2 was developed at INRA from a cross between *H. petiolaris* and HA61 [22].

The experiment was conducted at the research farm of the Faculty of Agriculture, Urmia University, located in Nazlu Region, Iran (latitude 45°37' N, longitude 5°32' E, altitude 1313 meters above sea level). The research farm is owned and managed by Urmia University, and no specific permits were required for access to this site as it is a designated agricultural research facility under the university's jurisdiction. The experimental work complied with all relevant institutional and national guidelines. The experiment employed a factorial design with two factors: salinity stress at two levels (control and stress conditions with electrical conductivity (EC) of 2 dS/m and 8 dS/m, respectively) and 96 inbred lines. The design was a completely randomized design (CRD) with three replications.

For each inbred line, six pots (diameter 26 cm, height 25 cm) were prepared and filled with a mixture of field soil and peat moss in a 3:1 ratio. The physical and chemical properties of the soil are presented in Table 1. The characteristics of the irrigation water used for both control and salinity stress treatments are shown in Table 2.

Salinity stress was induced by dissolving NaCl in water to achieve an EC of 8 dS/m. Specifically, 9.5 grams of NaCl were dissolved in 500 mL of water and applied to the pots after the seedlings reached the 8-leaf stage. To minimize plant shock, the saline solution was applied in two doses of 250 mL each, in the morning and afternoon. The control group was irrigated with

**Table 1. Physical and chemical properties of the soil used in the pots.**

| Parameter | Value |
| --- | --- |
| Soil texture | Loamy |
| Organic matter (%) | 1.8 |
| pH | 7.2 |
| EC (dS/m) | 1.5 |
| Total nitrogen (%) | 0.15 |
| Available phosphorus (mg/kg) | 12.5 |
| Available potassium (mg/kg) | 220 |

**Table 2. Chemical characteristics of the irrigation water.**

| Parameter | Control (EC = 2 dS/m) | Salinity Stress (EC = 8 dS/m) |
|---|---|---|
| pH | 7.0 | 7.0 |
| $Na^+$ (meq/L) | 5.0 | 20.0 |
| $Cl^-$ (meq/L) | 5.0 | 20.0 |
| $Ca^{2+} + Mg^{2+}$ (meq/L) | 2.0 | 2.0 |
| $SO_4^{2-}$ (meq/L) | 1.5 | 1.5 |
| $HCO_3^-$ (meq/L) | 1.0 | 1.0 |

water at an EC of 2 dS/m. Throughout the experiment, soil salinity in each pot was monitored using an electrical conductivity meter.

The pots were irrigated using a drip irrigation system, and fertilization was performed periodically during vegetative growth using a 20-20-20 (N-P-K) fertilizer. After the flowering stage, various traits were measured, including grain yield (GY, grams per plant), hundred-seed weight (HSW, grams), number of leaves (LN, counting from the first true leaf to the last fully developed leaf), leaf length (LL, cm), and leaf width (LW, cm). Additional measured traits were petiole length (PL, cm; average of upper, middle, and lower leaves), stem diameter (SD, cm), plant height (PH, cm), head diameter (HD, cm), head dry weight (HDW, grams), days to flowering (DF), and dry matter content.

A total of 288 data samples were collected (96 inbred lines × 3 replications). The dataset was divided into training and testing subsets using a 75:25 ratio, resulting in 216 samples for training and 72 samples for testing under both normal and salinity stress conditions.

## 2.2. Artificial neural networks

The architecture of an artificial neural network (ANN) typically includes an input layer, one or more hidden layers, and an output layer. In this study, the ANN model was developed using a three-layer feedforward network trained with the Levenberg-Marquardt backpropagation algorithm, known for its speed and reliability due to being a second-order nonlinear optimization technique [23–25]. The network's input layer consisted of ten neurons corresponding to the ten measured morphological traits. The number of neurons in the hidden layer was determined through trial and error, ranging from 1 to 10 neurons, to identify the optimal network structure that minimizes prediction error.

Various activation functions were tested for both hidden and output layers, including sigmoid, tangent sigmoid, and linear functions. The network weights and biases were adjusted to minimize the mean squared error (MSE) between observed and predicted grain yield values. Training was conducted until the MSE fell below a threshold of 0.001 or a maximum of 1,000 epochs was reached, serving as the convergence criterion. The ANN was implemented using MATLAB software, and the network structure is illustrated in Fig 1 [26].

Accordingly, after choosing the number of layers and the number of units in each layer, the weights and thresholds of the network should be adjusted to minimize the prediction error generated by the network [27]. Various activation functions were employed for both the hidden and output layers, including sigmoid logarithm, tangent sigmoid, and linear. The number of neurons in each hidden layer was determined via trial and error. Most previous studies have widely used the error test method to ascertain the ideal number of neurons in an ANN's hidden layer, e.g., [28–32]. The program code of the ANN was written by utilizing the MATLAB programming language. The structure of the artificial neural network for predicting sunflower seed yield is shown in Fig 1.

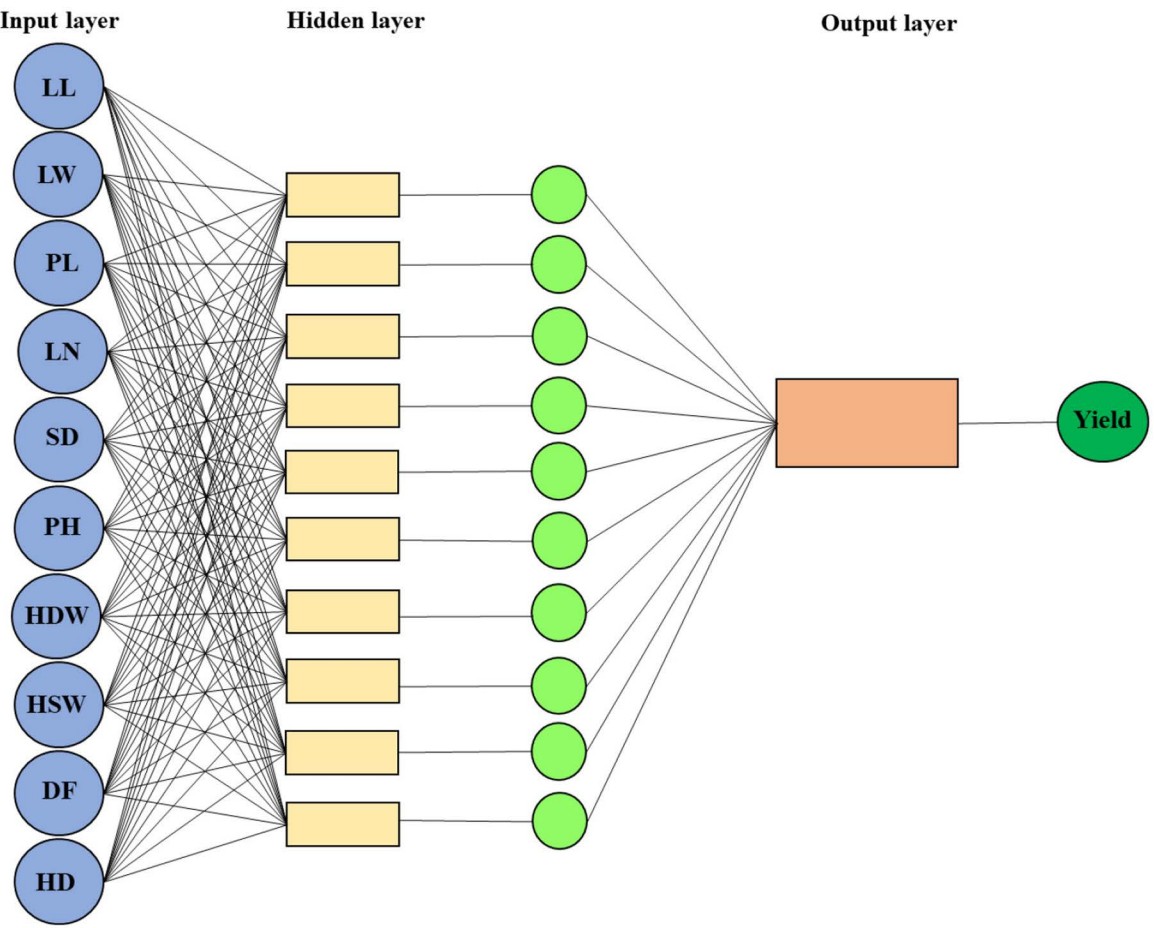

**Fig 1. Structure of artificial neural network for sunflower grain yield prediction.** LL: Leaf length; LW: Leaf width; PL: Petiole length; LN: Leaf number; SD: Stem diameter; PH: Plant height; HDW: Head dried weight; HSW: Hundred-seed weight; DF: Date to flowering; HD: Head diameter.

### 2.3. Adaptive neuro-fuzzy inference system

An adaptive neuro-fuzzy inference system (ANFIS) combines the learning capabilities of neural networks with the reasoning capabilities of fuzzy logic systems [20]. ANFIS can model complex nonlinear relationships by using fuzzy if-then rules and membership functions. In this study, the Sugeno-type fuzzy inference system was employed due to its effectiveness in handling continuous output variables [33].

The ANFIS model was developed using the subtractive clustering (SC) method to determine the optimal number of fuzzy rules and membership functions. The SC method considers each data point as a potential cluster center, which reduces computational complexity [34]. The radius of influence (RADII) parameter in SC was adjusted to control the granularity of the fuzzy partitions, with optimal values determined through experimentation.

Training of the ANFIS model continued until the MSE fell below 0.001 or a maximum of 1,000 epochs was reached, similar to the ANN model. MATLAB's Fuzzy Logic Toolbox was used to implement the ANFIS model.

Every fuzzy system includes three main parts: fuzzifying the data by defining the membership function, creating a connection between the input and output by means of a series

of rules (if-then), and gathering the results of the system and non-fuzzification. Fuzzy logic features are used to increase the performance of ANFIS (e.g., IF-THEN rules to estimate a non-linear function included in the modeling procedure). The IF part (antecedent) is fuzzy, while the THEN (consequent) part is an explicit function of an antecedent variable (typically, a linear equation).

$$Rule\ 1:\ If\ x\ is\ A_1\ and\ y\ is\ B_1,\ then,\ f_1 = p_1x + q_1y + r_1 \tag{1}$$

$$Rule\ 2:\ If\ x\ is\ A_2\ and\ y\ is\ B_2,\ then,\ f_1 = p_2x + q_2y + r_2 \tag{2}$$

where $A_1(LOW)$, $A_2(LOW)$, as well as $B_1(HIGH)$ and $B_2(MEDIUM)$ are the membership functions (MFs) for inputs $x$(CD) and $y$(PH), respectively. The ANFIS architecture is depicted in Fig 2.

The ANFIS subtractive clustering (ANFIS-SC), an extended model of the mountain clustering method, is obtained from the combination of ANFIS and SC. This model was proposed by Yager and Filev [35], where each data point (not a grid point) is considered a potential cluster center. Using the ANFIS-SC method has two main advantages. The quantity of effective "grid points" to be assessed is equivalent to the total number of data points independent of the problem's dimensionality. In addition, the technique obviates the requirement for determining mesh resolution, where the trade-off between accuracy and computational complexity must be considered.

The ANFIS-SC technique has expanded the criterion of the mountain method concerning the acceptance and rejection of cluster centers [36–38]. Two discrete program codes, comprising the fuzzy toolbox, were written using the MATLAB programming language for SC simulations.

## 2.4. Gene expression programming

Genetic programming (GP), which was first introduced by Cramer [39] and expanded by Koza [40], is considered one of the evolutionary algorithms and a subset of random search methods. GEP is a genetic algorithm (GA) and GP, and the difference between these three is like people [21]. In the GA, the nature of people acts as linear marked rows as bit strings with fixed length (chromosome) and based on the system of binary digits. In the GP, people are nonlinear with distinctive lengths and shapes and within the frame of parse trees. Moreover, in GEP, people are coded as checked lines with settled length (chromosome), then represented non-linearly

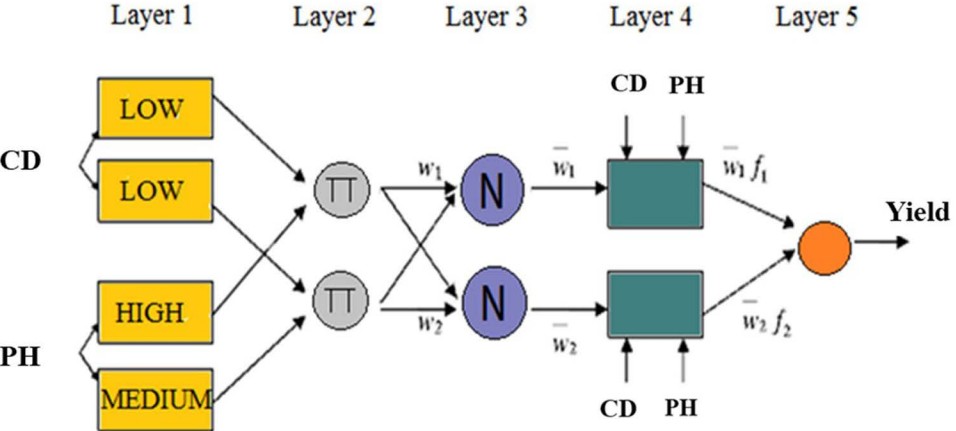

**Fig 2. Structure ANFIS model based on two input parameters (for the sample) to predict sunflower seed yield.**

and with different shapes and sizes by the expression tree [21]. The formation of the structure of chromosomes causes the formation of genes and the creation of the expression tree. Its head and tail regions regulate the structure of each gene. Therefore, there are two expressions in GEP, including the gene and expression tree. In the tree structure, each branch consists of a set of terminals (independent variables of the problem and system state variables, as well as constant and random numbers) and a set of functions (an arithmetic operator and the main trigonometric functions); these functions are located at the head of the gene [41], but only the main functions are placed in the terminal part [42]. The GEP models various genetic operators for functions, and the terminal set is used to construct a chromosome. An assortment of mathematical functions was employed to assess the process of modeling the sunflower grain yield.

$$(lnx,\ e^x,\ x^2,\ x^3,\ \sqrt{x},\ \sqrt[3]{x},\sin x,\cos x, arc\ tan(x))$$

This study's terminal set includes morphological variables such as LL, LW, PL, LN, SD, PH, HDW, HSW, DF, and HD. A powerful soft computing package called GeneXpro Tools 4.0 was employed to solve the problem of GYP predicting. In the present study, the count of chromosomes was established at 30. Further, the head's length, h = 7, and the number of three genes per chromosome were based on the GeneXpro default function set and utilized in implementing the GEP technique. The linking of the sub-trees was accomplished through the process of addition. Table 3 presents the values of the applied GEP model operators as scrutinized in this investigation. The main steps of this research to predict the sunflower seed yield using ANN, ANFIS, and GEP models are illustrated in Fig 3.

## 2.5. Evaluation of model performance

The present investigation entails an evaluation of model performance by utilizing statistical metrics, including the correlation coefficient (R), root means squared error (RMSE), and mean absolute error (MAE) (Eqs 3–5). The R, RMSE, and MAE are expressed as follows:

$$R = \left[ \frac{\sum_{i=1}^{N}\left(Y_{io} - \bar{Y}_o\right)\left(Y_{ie} - \bar{Y}_e\right)}{\sqrt{\sum_{i=1}^{N}(Y_{io} - \bar{Y}_o)^2 \sum_{i=1}^{N}(Y_{ie} - \bar{Y}_e)^2}} \right] \tag{3}$$

Table 3. The parameters of the GEP method utilized in the present investigation.

| General | |
|---|---|
| Linking function | Addition |
| Fitness function error type | RRSE |
| Penalizing tool | Parsimony pressure |
| **Genetic operator** | |
| Mutation rate | 0.044 |
| Inversion rate | 0.1 |
| IS transposition rate | 0.1 |
| RIS transposition rate | 0.1 |
| One-point recombination rate | 0.3 |
| Two-point recombination rate | 0.3 |
| Gene recombination rate | 0.1 |
| Gene transposition rate | 0.1 |

GEP: Gene expression programming; RRSE: Root relative squared error.

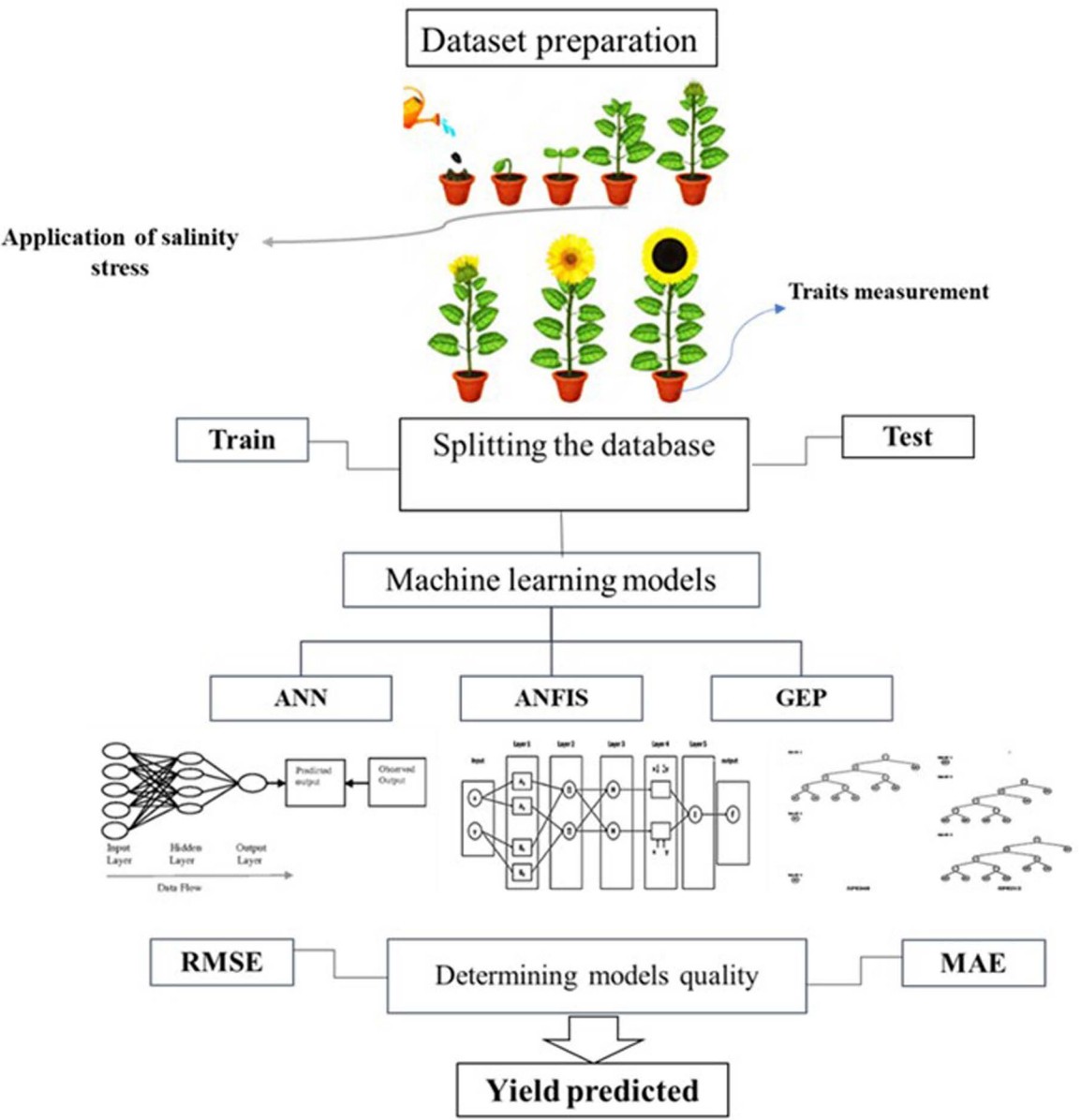

**Fig 3. Sunflower seed yield prediction steps using ANN, ANFIS, and GEP models.**

$$\text{RMSE} = \sqrt{\frac{1}{N}\sum_{i=1}^{N}\left(Y_{io} - Y_{ie}\right)^2} \tag{4}$$

$$MAE = \frac{1}{N}\sum_{i=1}^{N}\left|Y_{io} - Y_{ie}\right| \tag{5}$$

where $\overline{Y}_o$ and $\overline{Y}_e$ denote the average of the observed and estimated average yield sunflower values. Furthermore, $Y_{io}$ and $Y_{ie}$ represent the observed and estimated yield grain sunflower values, respectively, and N is the total number of data sets considered in this investigation. The correlation coefficient (R) is a statistical measure that determines the strength and

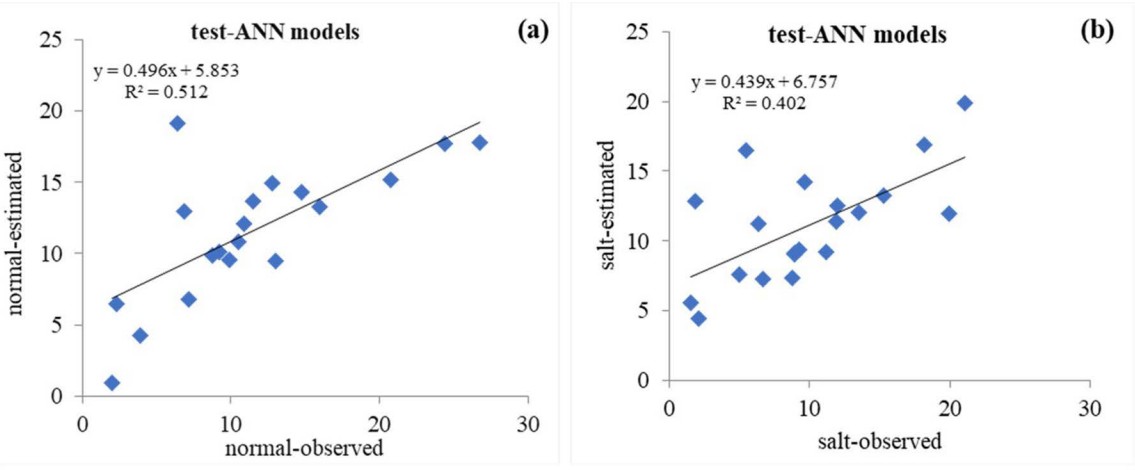

**Fig 4. Scatter plot of the observed (actual) vs. predicted values of sunflower grain yield with the ANN model in the test phase: (a) Normal conditions and (b) Salinity stress conditions.**

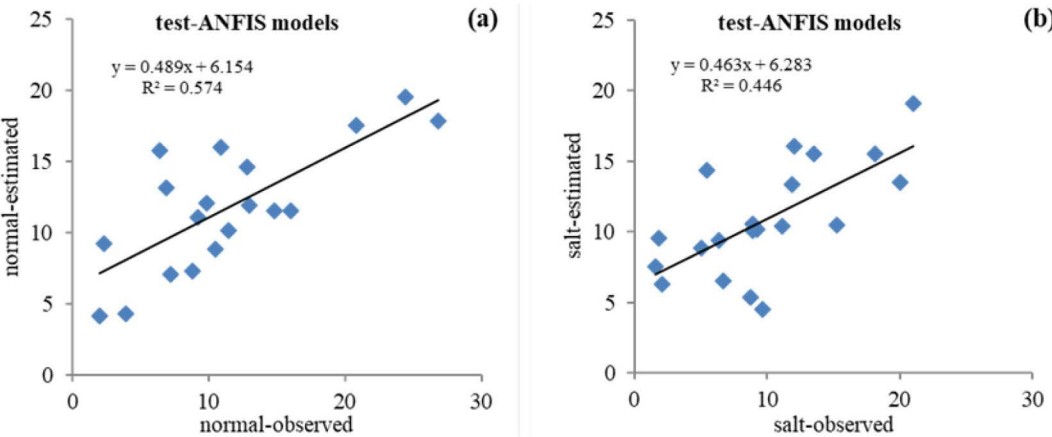

**Fig 5. Scatter plot of the observed (actual) vs. predicted values of sunflower grain yield with the ANFIS model in the test stage: (a) Normal conditions and (b) Salinity stress conditions.**

direction of the linear association between variables. The RMSE shows the goodness of fit relevant to high values, whereas the MAE is a more impartial evaluation of the goodness of fit at the moderate values range [36]. In summary, the optimal performance of the models is achieved when the values of R and RMSE are close to 1 and 0, respectively. To model the yield of sunflower seeds, the data were classified into training and test parts, with a ratio of 0.75 and 0.25 under normal conditions and salinity stress.

## 2.6. Prediction uncertainty analysis

Prediction uncertainty arises from various sources, including measurement errors, model structure limitations, and variability in environmental conditions. To assess the uncertainty in model predictions, residual analysis was conducted by examining the differences between observed and predicted grain yield values. The models' ability to handle uncertainty was evaluated based on their robustness and consistency across different conditions.

## 3. Results and discussion

The results of the descriptive statistics for the studied traits under normal and salt stress conditions in the population of recombinant inbred lines are presented in Table 4.

A hidden layer and the number of hidden nodes between 1 and 10 neurons were determined by trial and error and used in the ANN. An ANN with two and three hidden nodes has the highest coefficient of determination ($R^2$) and the lowest RMSE for normal and salinity stress conditions, respectively. There is no particular rule for determining the RADII value of ANFIS-SC models. The RADII values of the ANFIS-SC model were obtained for the optimal model of 0.32 and 0.26. The main advantage of GEP over the other data-driven techniques (e.g., ANFIS and ANN) is in generating explicit relationship formulas.

### 3.1. ANN models

The best topology of sunflower seed yield prediction with ANN has an input layer with 11 input variables (LL, LW, PL, LN, SD, PH, CDW, HSW, GYP, DF, and CD), a hidden layer with ten neurons and an output layer with one parameter (i.e., 11-10-1 structure) (Fig 1). The values of the accuracy evaluation parameters of the ANN model in the test stage are r = 0.716, RMSE = 4.638, and MAE = 3.210 under normal conditions, as well as r = 0.634, RMSE = 4.661, and MAE = 2.989 under salinity stress conditions (Table 5). Fig 4a and 4b displays scatter diagrams to examine the matching between the observed (actual) and predicted values for the sunflower grain yield by the ANN model in two conditions of normal ($R^2 = 0.512$) and salinity stress ($R^2 = 0.402$) in the test data set. Achieving a simple model with the least number of layers and hidden neurons and the most accurate values for the output variable (yield) is one of the main goals of ANN modeling studies [43]. The number of hidden layers in ANNs is influenced by the problem's complexity and the network's application. In determining the optimal structure of the ANN model, it is difficult to choose the appropriate number of neurons (nodes) in each hidden layer, and it is typically determined by trial and error; one or two hidden layers are often useful for the majority of problems [16]. A neural network with one hidden layer can approximate any continuous function if using sufficient connection weights [44]. Moradi, Bahmanyar [45] employed 10 neurons in the hidden layer and Levenberg-Marquardt, Logsig and Tansig inverse transfer functions for the hidden and output layer

**Table 4. Descriptive statistics for agricultural traits measured in the population of inbred sunflower recombinant lines.**

| Parameters | Normal | | | | Salt | | | |
|---|---|---|---|---|---|---|---|---|
| | Min. | Max. | Mean | Std. | Min. | Max. | Mean | Std. |
| LL | 8.11 | 20.39 | 11.28 | 1.60 | 7.67 | 32.61 | 11.66 | 3.14 |
| LW | 5.22 | 12.31 | 8.37 | 1.32 | 5.78 | 20.83 | 8.54 | 2.25 |
| PL | 4.17 | 10.50 | 7.02 | 1.16 | 3.61 | 10.83 | 7.08 | 1.19 |
| LN | 18.33 | 31.67 | 23.73 | 2.50 | 18.00 | 29.33 | 22.97 | 2.37 |
| SD | 3.17 | 5.50 | 4.46 | 0.45 | 3.17 | 5.33 | 4.29 | 0.44 |
| PH | 49.33 | 122.00 | 92.84 | 12.27 | 62.67 | 115.50 | 89.63 | 10.19 |
| HDW | 4.42 | 29.23 | 15.19 | 5.21 | 4.45 | 28.62 | 13.82 | 4.81 |
| HSW | 2.40 | 16.06 | 7.68 | 2.73 | 1.40 | 15.83 | 7.50 | 2.64 |
| DF | 67.33 | 96.50 | 77.50 | 4.80 | 67.33 | 92.67 | 75.76 | 3.76 |
| HD | 4.25 | 15.63 | 9.80 | 2.35 | 4.63 | 16.83 | 9.60 | 2.17 |
| GY | 0.63 | 37.57 | 12.08 | 7.25 | 0.59 | 28.74 | 10.74 | 6.26 |

Min: Minimum; Max: Maximum; Std.: Standard deviation; LL: Leaf length; LW: Leaf width; PL: Petiole length; LN: Leaf number; SD: Stem diameter; PH: Plant height; HDW: Head dried weight; HSW: 100 seeds weight; DF: Date to flowering; HD: Head diameter; GY: Grain yield.

**Table 5. Evaluating the efficacy of three models (ANN, ANFIS, and GEP) to predict sunflower grain yield under normal and salt stress.**

| Conditions | Test | | | | Train | | |
|---|---|---|---|---|---|---|---|
| | Model | $R^2$ | RMSE (Kg ha$^{-1}$) | MAE (Kg ha$^{-1}$) | $R^2$ | RMSE (Kg ha$^{-1}$) | MAE (Kg ha$^{-1}$) |
| Normal | GEP | 0.803 | 4.115 | 3.177 | 0.817 | 4.24 | 3.154 |
| | ANFIS | 0.758 | 4.403 | 3.491 | 0.784 | 4.566 | 3.445 |
| | ANN | 0.716 | 4.638 | 3.21 | 0.814 | 4.265 | 2.928 |
| Salt stress | GEP | 0.743 | 4.022 | 2.803 | 0.776 | 4.042 | 3.076 |
| | ANFIS | 0.668 | 4.238 | 3.516 | 0.745 | 4.264 | 3.037 |
| | ANN | 0.634 | 4.461 | 2.989 | 0.769 | 4.107 | 3.059 |

ANN: Artificial neural network; ANFIS: Adaptive neural inference system; GEP: Gene expression programming; RMSE: Root means squared error; MAE: Mean absolute error; $R^2$: Coefficient of determination.

algorithm as the best parameters of an ANN to model and optimize the extraction of anethole from fennel seeds with the help of ultrasound. Likewise, Sabzi-Nojadeh, Niedbała [46] reported the Sigmoid Axon transfer function, Levenberg-Marquardt, Momentum and Conjugate Gradient learning algorithm with the 11-10-1 structure as the best parameters in the ANN model for predicting fennel trans-anethole yield percentage. Previously, linear models, including regression models, were used to predict product performance. But, in MLR models, it is almost impossible to find the best model that gives estimates in accordance with the real data. Accordingly, biologists rarely use their regression model for prediction and attempt to perform a regression analysis to explain the effectiveness of independent variables on dependent variables [47]. Many studies introduce approaches that consider nonlinear relationships such as the ANN to be more accurate than the other methods because these methods, compared to linear methods, can better predict yield characteristics [48–51].

### 3.2. ANFIS and GEP models

The ANFIS model was evaluated to predict the sunflower grain yield. ANFIS is used in complex systems for modeling, control, or parameter estimation [52]. The results of investigating the accuracy of grain yield prediction with the ANFIS model in the test phase were $R^2 = 0.758$, RMSE = 4.403, and MAE = 3.491 under normal conditions, as well as $R^2 = 0.668$, RMSE = 4.238, and MAE = 3.516 under salinity stress conditions (Table 5). Fig 5a and 5b depict the quality of agreement between actual and predicted values for the grain yield in normal and salt stress conditions, respectively. The results confirmed the ability to predict sunflower grain yield with the ANFIS model. The details of the parameters used in the GEP model are provided in Table 3. The evaluation results of the accuracy of grain yield prediction with the GEP model in the testing phase were $R^2 = 0.803$, RMSE = 4.115, and MAE = 3.177, as well as $R^2 = 0.743$, RMSE = 4.022, and MAE = 2.803 under normal and salt stress conditions, respectively (Table 5). Fig 6a and 6b show the scatter diagram of actual versus predicted values for the grain yield with the GEP model in normal conditions and salt stress, respectively. The match's quality demonstrates the GEP model's high power in predicting the yield of sunflower grains under normal conditions and salinity stress.

To reflect the prediction accuracy of the developed models, Taylor and Violin charts and the comparison chart of the evaluation statistics of grain yield prediction accuracy have been drawn in the testing stage with ANN, ANFIS, and GEP models under normal and salt stress conditions (Figs 7–9, respectively). The Taylor diagram illustrates the degree of matching between observed and predicted data (yield) through a combination of $R^2$, RMSE, and

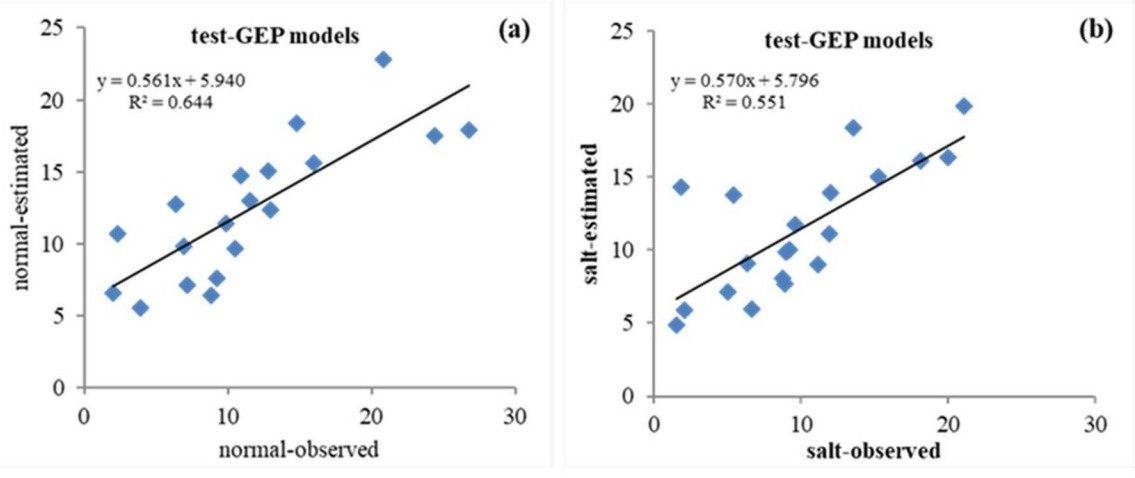

**Fig 6. Scatter plot of the observed (actual) vs. predicted values of sunflower grain yield with the GEP model in the test stage: (a) Normal conditions and (b) Salinity stress conditions.**

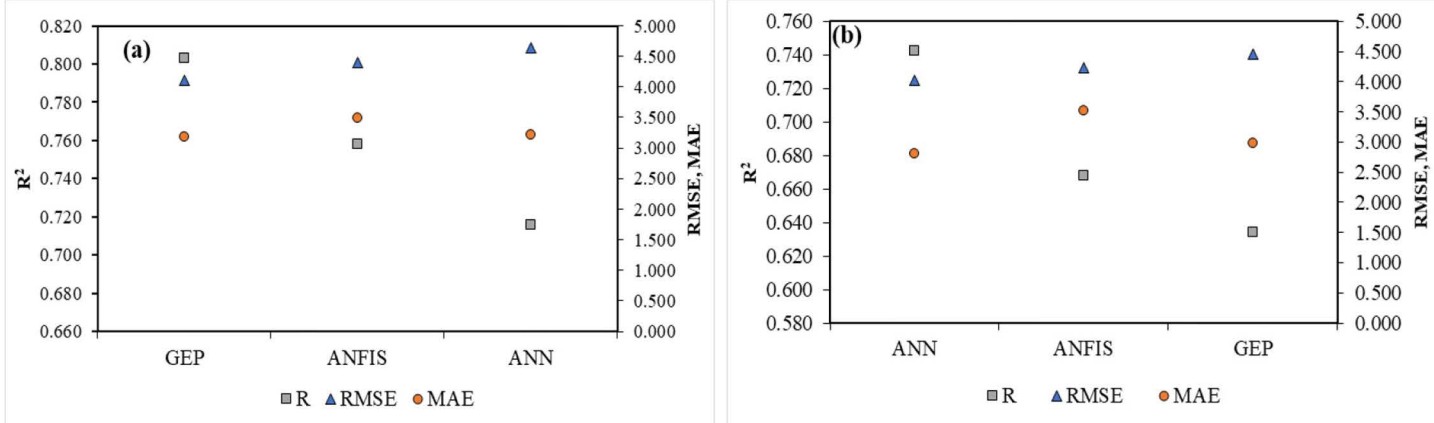

**Fig 7. Comparison of the accuracy evaluation statistics of models (ANN, ANFIS, and GEP) to predict the yield of sunflower grains in the test stage: (a) Normal conditions and (b) Salt stress conditions.**

standard deviation (SD) [53]. The black circle shows the observed data and the colored circles display the modeling results with the examined models. RMSE is proportional to the distance between the colored and black circles, and the radial axis (its radial distance from the origin) represents SDs. In contrast, the angular axis indicates the correlation coefficients [54]. The Taylor diagram depicts the accuracy of prediction models based on the distance between the black (observed) and colored (estimated) circles. Fig 8 shows the degree of agreement between the observed and predicted values with ANN, ANFIS, and GEP models for the sunflower grain yield in each normal and salt stress condition. The SDs of ANN, ANFIS, and GEP models under normal conditions in the test phase are approximately 3, 3, and 3.6, respectively, while the observed SD was 6.5. Compared to ANN and ANFIS, the GEP model has a higher correlation coefficient. At the same time, a lower RMSE is introduced as the superior model (Fig 8a). In salinity stress conditions, the SDs of ANN, ANFIS, and GEP models were nearly 2.4, 1.1, and 3.1, respectively, while the observed SD was 5.5. The GEP model under salinity stress has a lower value of RMSE and SD.

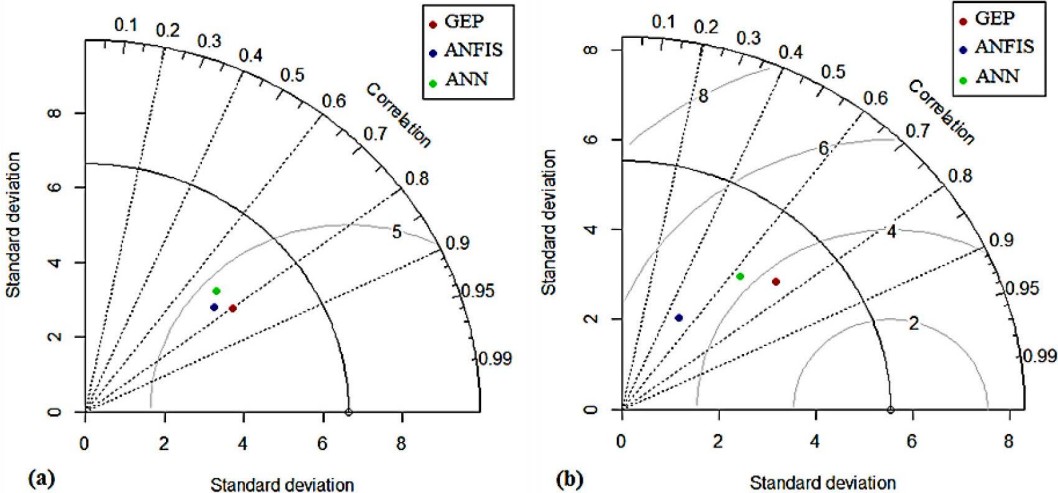

**Fig 8. Taylor diagrams to compare the performance of models (ANN, ANFIS, and GEP) to predict the yield of sunflower grains in the test stage: (a) Normal conditions and (b) Salt stress conditions.**

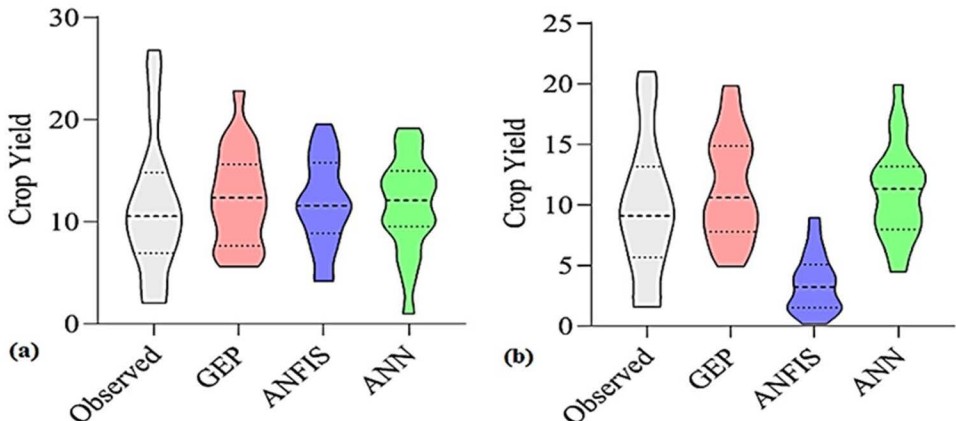

**Fig 9. Violin diagrams of models (ANN, ANFIS, and GEP) to predict the yield of sunflower grains in the test stage: (a) Normal conditions and (b) Salt stress conditions.**

In comparison, a higher value of the correlation coefficient and the results match the observations in the best way (Fig 8b). The high accuracy of this chart and its superiority are discernible in Fig 7, which illustrates the comparison of the accuracy evaluation statistics of the models (ANN, ANFIS, and GEP) in predicting the sunflower grain yield under both conditions in the test stage. This graph displays the superiority of the GEP model in predicting grain yield in normal and salt stress conditions, which has less error with higher $R^2$ than the other two models. The violin diagrams of the examined models for predicting the grain yield under normal and salinity stress conditions are depicted in Fig 9a and 9b. A violin plot consists of a box plot and a density chart, presenting a comprehensive visualization of the distribution and the peaks of the data. This combination of density tracing and box plot allows a quick and precise comparison between several distributions [55]. Based on the violin diagrams in normal conditions (Fig 9a), the ANFIS model is more similar to the real data set graph than the other models.

In contrast, in the salt stress condition (Fig 9b), the GEP model is more similar to the real data set's graph than the other models. Essentially, the statistical properties of the predicted yield values from ANFIS and GEP models closely mirror those of actual data; in other words, these two models are more successful in performance modeling than the ANN model. It should be noted that the graphical comparison of the obtained results is justified by using the statistical indicators presented in Table 5. As mentioned earlier, the current study investigated the ability of three ANN, ANFIS, and GEP models to predict the sunflower grain yield under normal conditions and salt stress. A comparison of accuracy evaluation results demonstrates that the GEP model has lower RMSE, MAE, and higher $R^2$ than the ANN and ANFIS models. Therefore, compared to the other investigated models, it is superior in predicting the sunflower grain yield. In addition, by comparing ANFIS and ANN models, it is clear that the ANFIS model can predict yields with higher correlation coefficients and smaller RMSE values than ANNs. Based on the information, the ANN model performs less than the other methods in predicting the grain yield (Table 5). Figs 4 –6 depict the scatter plots of actual versus predicted values with the studied models at the test level under normal conditions and salt stress. The GEP model with $R^2$ equal to 0.644 and 0.551 in normal and salt stress conditions, respectively, provides the best results compared to the other models. Based on the RMSE criterion, the percentage improvement in the efficiency of the GEP model is 6.54% and 11.27% compared to the ANFIS and ANN models, respectively.

Furthermore, compared to the ANN, the efficiency improvement percentage of the ANFIS model is 50.06%. Based on Taylor diagrams, the GEP model is superior to the other studied models in terms of higher correlation coefficient and lower RMSE in both normal and salt stress conditions. One of the significant benefits of using ANNs is their flexibility and potential to model nonlinear relationships. The GEP is one of the most powerful machine coding resources for solving nonlinear problems [56]. The GEP nonlinear model has shown successful applications with high-performance accuracy [57]. One of the most important features of independent GEPs can be considered to solve the overfitting problem [58]. One of the important priorities of GEP is the automatic selection of effective parameters for modeling and presentation of the mathematical relationship governing the problem. In this study, mathematical relations extracted from GEP for normal and salinity conditions are presented in relations (6) and (7), respectively.

$$\begin{aligned}y = {}& \sin(LN) + HDWA \tan(A \tan(SD^2)) + HSW - Ln\left(d_7^4\right) \\ & + \cos(Ln(CD).\left(PH + HSW\right)^{1/2}) + \cos(PH - 0.41)\end{aligned} \tag{6}$$

$$\begin{aligned}y = {}& \cos(LL + HD) - \cos(HSW^{\frac{1}{3}} - \sin(HD)) + \sin[\sin(\cos(HSW) - A \tan(HSW)) \\ & - \frac{A \tan(CD + 3.15)}{A \tan(CDW)} + HD\, A \tan(HSW^{1/2}) + HSW^{1/2} \cos(LL)\end{aligned} \tag{7}$$

According to Eq. (6), obtained from the GEP method for predicting performance under normal conditions, six parameters, including LN, HDW, SD, HSW, HD, and PH, were entered into the model. However, under salinity stress conditions (Eq. 7), four LL, HD, HSW, and HDW parameters were included in the yield prediction equation. These parameters can be considered part of the attributes affecting performance and modeling by the GEP model. The importance of the parameters entered into the performance prediction model can be searched in the other research data. The reduction of LL in sunflowers under salinity stress conditions is a tolerance mechanism. Salinity stress affects sunflower performance through nutritional

imbalance and water deficit stress. To reduce nutrient imbalances and osmotic stress caused by soil salinity, sunflowers use mechanisms to reduce water loss while maximizing water absorption, including leaf area reduction and osmotic regulation [59,60]. The osmotic adjustment capacity allows the plant to continue its growth in saline conditions. The accumulation of ions (especially sodium) in leaf tissues is the primary mechanism of osmotic regulation [61]. Effective enhancement in the sunflower seed yield can be achieved through selection based on PH, HD, and HSW, as they exhibit a significant correlation with the sunflower seed yield [62]. Among the morpho-physiological traits, the relationship between traits such as PH and HD with the sunflower seed yield is positively correlated [63–66]. The weakness of ANN in performance prediction can be attributed to the unexplained and challenging behavior in the ANN network structure [19].

Additionally, it is believed that GEP performs better than ANN and ANFIS due to its ability to formulate relevant mathematical equations that streamline the estimation of output parameters [67]. The distinct structure of each model is the reason for the variation in their performance. The GEP approach, due to its utilization of gene and chromosome structure, and the ANFIS approach, through the amalgamation of fuzzy theory and neural network techniques, have demonstrated superior performance compared to ANNs [53]. Based on the literature review, there is no unique model superior to others in all cases, and the efficacy of different models may differ according to the conditions of each hopological system [57]. The findings from this research highlight the prowess of ANN models in accurately estimating the sunflower seed yield in normal and salinity conditions. The primary practical use of the results will be in predicting profitability before harvest. At the senior management level, awareness and forecasting of the amount of agricultural production can be used in pricing and determining the import and export of products. Therefore, the predictive models of plant yield have the potential to give rise to foretelling tools that play a vital role in advanced farming practices, serving as the primary component of frameworks designed to support decision-making processes [68]. Estimating the yield of sunflower seeds in saline soils is necessary to perform technical-economic analysis to support the decisions of small farmers. Alternative techniques should be developed for grain yield estimation. A key motivation for the use of machine learning in plant breeding is the ever-increasing volume of data generated by high-throughput phenotyping and genotyping, complemented by rich environmental information from weather stations and satellites [69]. The analysis of high-volume data necessitates a shift in analytical mindset as conventional statistical techniques may not be appropriate for extensive data sets. Instead, novel algorithms can aid in the extraction of valuable new patterns from the data. Awareness and comprehension of various kinds of machine learning techniques enable breeders to choose the right method for a given task and parameterize them with their knowledge [70,71].

## 4. Conclusion

This study used the ANN, ANFIS, and GEP models to predict the sunflower seed yield under normal conditions and salt stress. The efficiency of ANN, ANFIS, and GEP models in predicting performance was evaluated based on statistical indicators such as $R^2$, MAE, and RMSE. It was concluded that the ANFIS model performed better and more reliably than the ANN and GEP model compared to ANN and ANFIS models in predicting the sunflower seed yield. In the GEP model, selecting effective parameters for modeling and presenting the problem's mathematical relationship is among its main advantages; thus, it can be considered a highly good alternative to linear modeling and other non-linear models. Therefore, this model can be used to predict the seed yield in sunflowers with high accuracy, less cost, and more time-saving.

## Author contributions

**Conceptualization:** Sanaz Khalifani.

**Data curation:** Majid Montaseri.

**Formal analysis:** Majid Montaseri.

**Methodology:** Sanaz Khalifani.

**Project administration:** Reza Darvishzadeh, Mojtaba Kordrostami.

**Resources:** Sarvin Zaman Zad Ghavidel.

**Software:** Sarvin Zaman Zad Ghavidel.

**Supervision:** Reza Darvishzadeh, Sarvin Zaman Zad Ghavidel, Mojtaba Kordrostami.

**Validation:** Majid Montaseri, Hamid Hatami Maleki.

**Writing – original draft:** Majid Montaseri, Hamid Hatami Maleki.

**Writing – review & editing:** Reza Darvishzadeh, Hamid Hatami Maleki.

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
