## [Decision Letter · Decision Letter 0]

20 Sep 2024

PONE-D-24-00132

Advanced Computational Approaches for Predicting Sunflower Yield: Insights from ANN, ANFIS, and GEP in Normal and Salinity Stress Environments

PLOS ONE

Dear Dr. Kordrostami,

Thank you for submitting your manuscript to PLOS ONE. After careful consideration, we have decided that your manuscript does not meet our criteria for publication and must therefore be rejected.

Specifically:

Unfortunately, in the opinion of the reviewers, the article is not acceptable.

I am sorry that we cannot be more positive on this occasion, but hope that you appreciate the reasons for this decision.

Kind regards,

Somayeh Soltani-Gerdefaramarzi, Ph. D.

Academic Editor

PLOS ONE

Reviewers' comments:

Reviewer's Responses to Questions

**Comments to the Author**

1. Is the manuscript technically sound, and do the data support the conclusions?

Reviewer #1: Yes

Reviewer #2: No

2. Has the statistical analysis been performed appropriately and rigorously? 

Reviewer #1: Yes

Reviewer #2: Yes

3. Have the authors made all data underlying the findings in their manuscript fully available?

Reviewer #1: Yes

Reviewer #2: Yes

4. Is the manuscript presented in an intelligible fashion and written in standard English?

Reviewer #1: Yes

Reviewer #2: Yes

5. Review Comments to the Author

Reviewer #1: The objective of this research is to predict the sunflower grain yield in normal conditions and salinity stress using three modeling techniques such as an adaptive neural inference system, artificial neural networks, and gene

expression programming (GEP). The paper is well structured, and the results were well explained and elaborated.

Reviewer #2: I studied the manuscript carefully and I have some suggestions and questions regarding the article's improvement.

- Abstract: What data range did the authors use for this study? what is the scale of data?

- What were salt stress conditions? Explain in Abstract.

- How the authors recognized four parameters of LL, head diameter, HSW, and HDW are the most significant parameters? With what methods (sensitivity analysis, trial and error, etc)?

- In general, the Abstract section is incomplete and strongly suggest to revise and rewrite this section.

- There is a need to explain robustly what limitations and gaps were found in previous studies in yield prediction using different artificial intelligence models in which the author encouraged to develop current study.

- What are the advantages of ANFIS and GEP over ANN that need to be used for predicting grain yield?

- What was the EC (dS/m) value of the irrigation water used in the control and salinity pots? Please provide two Tables to show characteristics of irrigation water and soil.

- What is the convergence criterion of models?

- The prediction uncertainty is a critical issue. It deserves a discussion.

- How many data used for modeling? How did the authors divide the data into training and testing subsets, and what percentage of the data was allocated to each subset?

6. PLOS authors have the option to publish the peer review history of their article (what does this mean? ). If published, this will include your full peer review and any attached files.

**Do you want your identity to be public for this peer review?** For information about this choice, including consent withdrawal, please see our Privacy Policy .

Reviewer #1: **Yes: **

Reviewer #2: No

- - - - -

---

## [Author Response · Author response to Decision Letter 0]

9 Oct 2024

Dear Editorial office,

We are pleased to resubmit our revised manuscript entitled "Advanced Computational Approaches for Predicting Sunflower Yield: Insights from ANN, ANFIS, and GEP in Salinity Stress Environments" for consideration in PLOS ONE

We sincerely appreciate the valuable comments and suggestions provided by both reviewers, which have greatly helped us to improve the quality and clarity of our manuscript. We have carefully considered all the comments and have revised the manuscript accordingly.

Reviewer #1:

The objective of this research is to predict the sunflower grain yield in normal conditions and salinity stress using three modeling techniques such as an adaptive neural inference system, artificial neural networks, and gene expression programming (GEP). The paper is well structured, and the results were well explained and elaborated.

Response:

Thank you very much for your positive feedback on our manuscript. We are delighted to hear that you found our paper well-structured and that the results were clearly explained and elaborated. Your encouraging comments have motivated us to further refine the manuscript. In addition to addressing Reviewer #2's comments, we have made minor improvements throughout the manuscript to enhance clarity and readability.

Reviewer #2:

1. Abstract: What data range did the authors use for this study? What is the scale of data?

Response:

Thank you for pointing out the need to specify the data range and scale in the abstract. We have revised the abstract to include these details. Specifically, we have mentioned that the study involved a pot experiment with 96 inbred sunflower lines (generation six) derived from crossing two parent lines, conducted over a single growing season.

2. What were salt stress conditions? Explain in Abstract.

Response:

We appreciate your suggestion to include the salinity levels used in the study within the abstract. We have updated the abstract to specify that salinity stress was induced by applying irrigation water with electrical conductivity (EC) levels of 2 dS/m (control) and 8 dS/m (stress condition), using NaCl, applied after the seedlings reached the 8-leaf stage.

3. How did the authors recognize four parameters of LL, head diameter, HSW, and HDW as the most significant parameters? With what methods (sensitivity analysis, trial and error, etc)?

Response:

Thank you for highlighting the need to explain our method for identifying significant parameters. We have clarified in both the abstract and the Methods section that we used sensitivity analysis within the GEP model to identify leaf length (LL), head diameter (HD), hundred-seed weight (HSW), and head dry weight (HDW) as the most significant parameters influencing grain yield under salinity stress.

4. In general, the Abstract section is incomplete and strongly suggest to revise and rewrite this section.

Response:

We appreciate your feedback regarding the abstract. We have thoroughly revised and rewritten the abstract to ensure it includes all essential components: background, objectives, methods, key results, and conclusions.

5. There is a need to explain robustly what limitations and gaps were found in previous studies in yield prediction using different artificial intelligence models that encouraged the development of the current study.

Response:

Thank you for this important suggestion. In the Introduction, we have expanded the discussion to highlight the limitations of previous studies using ANN models, such as overfitting, lack of interpretability due to their "black-box" nature, and difficulties in capturing complex nonlinear interactions among variables. We emphasized how these limitations motivated us to explore ANFIS and GEP as alternative modeling techniques that offer improved interpretability and handling of nonlinear relationships.

6. What are the advantages of ANFIS and GEP over ANN that need to be used for predicting grain yield?

Response:

We agree that discussing the advantages of ANFIS and GEP is crucial. We have included a section in the Introduction that explains how ANFIS combines the learning capabilities of neural networks with the reasoning abilities of fuzzy logic, effectively handling uncertainty and complex nonlinear systems. GEP, on the other hand, evolves explicit mathematical expressions, enhancing interpretability and providing insights into underlying variable relationships. These characteristics make ANFIS and GEP promising alternatives to traditional ANN models.

7. What was the EC (dS/m) value of the irrigation water used in the control and salinity pots? Please provide two tables to show characteristics of irrigation water and soil.

Response:

Thank you for pointing out the need for clarity regarding the EC values and the characteristics of the irrigation water and soil. We have specified in the Methods section that the control group was irrigated with water at an EC of 2 dS/m, and the salinity stress group received water at an EC of 8 dS/m, achieved by dissolving NaCl. Additionally, we have included two tables presenting detailed chemical and physical properties of the soil (Table 1) and irrigation water (Table 2) used in the experiment.

8. What is the convergence criterion of models?

Response:

We appreciate your suggestion to include the convergence criteria for our models. In the Methods section, we have specified the convergence criteria for each modeling technique. For the ANN and ANFIS models, training continued until the mean squared error (MSE) decreased below 0.001 or a maximum of 1,000 epochs was reached. For the GEP model, convergence was based on achieving a stable fitness value or after 500 generations.

9. The prediction uncertainty is a critical issue. It deserves a discussion.

Response:

Thank you for highlighting the importance of addressing prediction uncertainty. We have added a subsection in the MM that explores the sources of prediction uncertainty, such as measurement errors and model limitations. We discuss how the GEP model's explicit mathematical equations allow for better understanding and quantification of uncertainty, and how ANFIS incorporates fuzzy logic to handle data uncertainty, improving prediction robustness.

10. How many data were used for modeling? How did the authors divide the data into training and testing subsets, and what percentage of the data was allocated to each subset?

Response:

We apologize for not providing this information earlier. In the Methods section, we have clarified that a total of 288 data samples were collected (96 inbred lines × 3 replicates). The dataset was divided into training and testing subsets using a 75:25 ratio, resulting in 216 samples for training and 72 samples for testing under both normal and salinity stress conditions.

We believe that these revisions have significantly improved the manuscript and have addressed all your concerns. We are grateful for your valuable feedback, which has enhanced the quality of our work.

Thank you once again for your time and consideration.

Sincerely,

Reza Darvishzadeh

M. Kordrostami

---

## [Decision Letter · Decision Letter 1]

22 Nov 2024

PONE-D-24-00132R1Advanced Computational Approaches for Predicting Sunflower Yield: Insights from ANN, ANFIS, and GEP in Normal and Salinity Stress EnvironmentsPLOS ONE

Dear Dr. Kordrostami,

Thank you for submitting your manuscript to PLOS ONE. After careful consideration, we feel that it has merit but does not fully meet PLOS ONE’s publication criteria as it currently stands. Therefore, we invite you to submit a revised version of the manuscript that addresses the points raised during the review process.

Dear authorsI believe the paper has the potential for significant improvement. Therefore, I kindly request the authors to revise the manuscript, addressing all the provided comments thoroughly. Once revised, please resubmit the paper for further evaluation.

We look forward to receiving your revised manuscript.

Kind regards,

Morteza Taki, Ph.D

Academic Editor

PLOS ONE

Journal Requirements:

4. We note that your Data Availability Statement is currently as follows: "All relevant data are within the manuscript and its Supporting Information files."

The values behind the means, standard deviations and other measures reported;

The values used to build graphs;

The points extracted from images for analysis.

6. Thank you for updating your data availability statement. You note that your data are available within the Supporting Information files, but no such files have been included with your submission. At this time we ask that you please upload your minimal data set as a Supporting Information file, or to a public repository such as Figshare or Dryad.

Please also ensure that when you upload your file you include separate captions for your supplementary files at the end of your manuscript.

As soon as you confirm the location of the data underlying your findings, we will be able to proceed with the review of your submission.

7. We note you have included more than one caption for Tables [1, 2, 3, 4, 5] . Please remove the second caption from the manuscript, and ensure that each table has only one caption provided in the manuscript file.

8. Please upload a Response to Reviewers letter which should include a point by point response to each of the points made by the Editor and / or Reviewers. (This should be uploaded as a 'Response to Reviewers' file type.)

Please follow this link for more information: http://blogs.PLOS.org/everyone/2011/05/10/how-to-submit-your-revised-manuscript/

Additional Editor Comments (if provided):

Reviewers' comments:

Reviewer's Responses to Questions

**Comments to the Author**

1. If the authors have adequately addressed your comments raised in a previous round of review and you feel that this manuscript is now acceptable for publication, you may indicate that here to bypass the “Comments to the Author” section, enter your conflict of interest statement in the “Confidential to Editor” section, and submit your "Accept" recommendation.

Reviewer #3: (No Response)

Reviewer #4: All comments have been addressed

2. Is the manuscript technically sound, and do the data support the conclusions?

Reviewer #3: Partly

Reviewer #4: No

3. Has the statistical analysis been performed appropriately and rigorously? 

Reviewer #3: No

Reviewer #4: No

4. Have the authors made all data underlying the findings in their manuscript fully available?

Reviewer #3: Yes

Reviewer #4: No

5. Is the manuscript presented in an intelligible fashion and written in standard English?

Reviewer #3: No

Reviewer #4: Yes

6. Review Comments to the Author

Reviewer #3: Dear authors

Please see the comments

1. Clarity on Experimental Design: Can you provide more details on the rationale behind choosing the specific salinity levels (2 dS/m and 8 dS/m) for the experiment?

2. Model Selection Justification: Why were ANN, ANFIS, and GEP specifically chosen for this study? Are there other models that could have been considered?

3. Data Preprocessing: How was the data preprocessed before being fed into the models? Were there any missing values or outliers, and how were they handled?

4. Model Training Details: Can you elaborate on the training process for each model? For instance, how were the hyperparameters tuned?

5. Validation Methods: What methods were used to validate the models? Were cross-validation techniques employed?

6. Sensitivity Analysis: The sensitivity analysis identified LL, head diameter, HSW, and HDW as significant parameters. Can you explain why these particular traits are more influential under salinity stress?

7. Comparison with Previous Studies: How do the results of this study compare with previous studies on sunflower yield prediction under salinity stress?

8. Interpretability of Models: While GEP provides explicit mathematical expressions, how interpretable are the ANN and ANFIS models? Can you provide examples of the rules generated by ANFIS?

9. Model Performance Metrics: The paper mentions R², RMSE, and MAE. Why were these specific metrics chosen, and how do they complement each other in evaluating model performance?

10. Environmental Conditions: Were there any other environmental factors (e.g., temperature, humidity) monitored during the experiment that could have influenced the results?

11. Generalizability of Results: How generalizable are the findings to other sunflower genotypes or different environmental conditions?

12. Practical Applications: Can you discuss the practical implications of using these models for farmers and breeders? How can they integrate these tools into their decision-making processes?

13. Limitations and Future Work: What are the main limitations of this study, and how could future research address them?

14. Statistical Significance: Were the differences in model performance statistically significant? If so, what statistical tests were used to determine this?

15. Graphical Representations: The paper includes scatter diagrams and Taylor diagrams. Can you provide more interpretation of these figures and how they support the findings?

16. How could you clarify the results of ANN models? I mean you should use some cross validation methods such as K-Fold.

Reviewer #4: After reviewing the authors' responses and the revised version of the manuscript, I must inform you that, in its current form, the article does not meet the necessary criteria for acceptance.

7. PLOS authors have the option to publish the peer review history of their article (what does this mean? ). If published, this will include your full peer review and any attached files.

**Do you want your identity to be public for this peer review?** For information about this choice, including consent withdrawal, please see our Privacy Policy .

Reviewer #3: No

Reviewer #4: No

---

## [Author Response · Author response to Decision Letter 1]

17 Jan 2025

January 3, 2025

Dear Editor-in-Chief,

I would appreciate it if you look into our revised manuscript entitled “Advanced Computational Approaches for Predicting Sunflower Yield: Insights from ANN, ANFIS, and GEP in Normal and Salinity Stress Environments”. As far as the referee’s propositions are concerned, all modifications have been applied and highlighted for your consideration. The manuscript was critically revised by an English-speaking native for spelling and grammar. We the thank referees for their valuable recommendations and hope that they will be satisfied with the revised manuscript.

Sincerely yours

Corresponding author

Response to reviewer comments:

The answer to the comments of the first referee

It should be noted that the suggestions of the first respected referee are highlighted in green in the text.

Answer to first question

Rationale for Selecting Salinity Levels (2 and 8 dS/m)

Practical agricultural considerations and the physiological responses of sunflower plants to salinity stress guided the selection of salinity levels at 2 and 8 dS/m. The following key points explain the rationale behind these choices:

1. Baseline Control (2 dS/m): A salinity level of 2 dS/m was chosen to represent the normal irrigation water salinity in the study region (Nazlou, Iran). This level serves as the control, reflecting standard growth conditions. This baseline aligns with typical irrigation practices in the experimental area, enhancing the study's relevance to local farming conditions.

2. Stress Level (8 dS/m): The 8 dS/m salinity level was selected to simulate moderate salinity stress commonly encountered in arid and semi-arid regions, where sunflower cultivation is prevalent. According to Maas and Hoffman (1977), sunflower is categorized as a relatively sensitive crop, with yield reductions starting at salinity levels above 4 dS/m. The 8 dS/m level was chosen to induce measurable stress responses without causing complete plant failure. This level is consistent with findings by Katerji et al. (2000), who observed significant physiological and yield responses in sunflower at salinity levels ranging from 6 to 10 dS/m. Thus, 8 dS/m represent a practical threshold for moderate stress, enabling meaningful comparisons of morphological traits and yield between normal and stressed conditions.

3. Relevance to local agricultural practices: The selected salinity levels aim to impose stress that does not entirely incapacitate the plants but allows for measurable changes in performance. Farmers in the region often face irrigation water salinity levels between 2 and 8 dS/m due to factors such as soil salinity and limited access to freshwater resources. Therefore, these levels enhance the study's relevance to real-world scenarios and support the development of predictive models that can guide irrigation strategies and crop management under varying salinity conditions.

By selecting these salinity levels, the study ensures both scientific validity and practical applicability, facilitating the development of tools to predict and manage sunflower production under diverse environmental conditions.

Answer to second question:

Justification for Model Selection: Why ANN, ANFIS, and GEP?

The selection of Artificial Neural Networks (ANN), Adaptive Neuro-Fuzzy Inference Systems (ANFIS), and Gene Expression Programming (GEP) was underpinned by their demonstrated ability to model complex and nonlinear relationships in agricultural and environmental datasets, particularly under abiotic stresses like salinity. A detailed explanation for their inclusion is provided below:

1. Artificial Neural Networks (ANN):

• Flexible Nonlinear Modeling: ANNs are well-suited for capturing complex nonlinear relationships between input variables (morphological features) and outputs (grain yield). Their ability to learn from data without rigid assumptions makes them powerful for yield prediction.

• Proven Success in Agriculture: Previous research has shown that ANNs can outperform traditional linear regression models in prediction accuracy when dealing with agricultural data and yield estimation.

• Adaptability: ANNs can be easily adjusted in terms of architecture (number of layers and neurons), learning algorithms, and activation functions, allowing for optimal performance on specific datasets.

2. Adaptive Neuro-Fuzzy Inference Systems (ANFIS): ANFIS combines the learning capabilities of ANNs with the interpretability of fuzzy logic. This made it a suitable choice due to its ability to model uncertain and imprecise data, a common challenge in agricultural studies. ANFIS can emulate human-like reasoning and provide rule-based outputs that are easily interpretable, beneficial for translating findings into practical agricultural practices.

3. Gene Expression Programming (GEP): GEP was included due to its strength in symbolic regression and its ability to evolve models that explicitly explain the relationships between variables in mathematical form. This feature is crucial for generating predictive models that not only forecast outcomes but also provide insights into the underlying mechanisms of stress responses.

While ANN, ANFIS, and GEP were selected for their respective strengths, other modeling methods could have been considered, such as:

• Support Vector Machines (SVM)

• Random Forest (RF)

• Gradient Boosting Machines (GBM) and XGBoost

• Deep Learning (e.g., Convolutional Neural Networks - CNN, Recurrent Neural Networks - RNN): Deep learning models could be considered for handling large and complex datasets, especially when spatial or temporal dynamics are important.

Justification for not including other models: While models like SVM, RF, and GBM are powerful, the primary motivation for selecting ANN, ANFIS, and GEP was their combined ability to handle nonlinearity, uncertainty, and interpretability. This combination is critical in agricultural studies, where accurate prediction and understanding of underlying processes are vital. The selected models effectively meet the study's goals of predicting outcomes and elucidating plant responses to salinity stress.

Answer to third question

Data Preparation and Preprocessing

A dataset encompassing 104 sunflower recombinant inbred lines was evaluated under two normal and salinity stress conditions. After data collection, missing values within the dataset were identified. This process involved meticulously examining each trait to identify gaps in the data inputs. Subsequently, lines with a high proportion of missing values and data points identified as outliers and deemed measurement errors were removed from the dataset. This action was taken to preserve the model's integrity under normal and salinity stress conditions. Following this step, the mean values for all data points were calculated. Next, the dataset of mean values was divided into training and testing subsets using a 75:25 ratio, as previously described in the text. This division ensures that the models are trained solely on one dataset while being validated on an independent subset. This approach allows for an unbiased evaluation of model performance and enhances the accuracy of the generated predictions.

4-The fourth question

Model training details: Can you elaborate on the training process for each model?

1. Artificial Neural Networks (ANN)

• Training Process:

• Initialization: The ANN model begins with the establishment of its architecture, which includes an input layer corresponding to the 10 morphological traits, one or more hidden layers, and an output layer designed to predict grain yield.

• Data Preparation: The training dataset, consisting samples obtained from a 75:25 split, is utilized to fit the model.

• Hyperparameter Tuning

• Number of Hidden Layers and Neurons: An iterative trial-and-error methodology was employed to test various configurations of hidden layers, specifically ranging from 1 to 10 neurons per layer. Model performance was evaluated using mean squared error (MSE) until optimal configurations that minimized prediction error were identified.

• Activation Functions: Different activation functions—including sigmoid, hyperbolic tangent, and linear—were explored for both hidden and output layers to effectively capture non-linear relationships within the data.

• Training Algorithm: The Levenberg-Marquardt backpropagation algorithm was used for its efficiency in model training. This second-order optimization technique helps facilitate rapid convergence to a solution with minimal error.

• Training Epochs: The training procedure was designed to continue until the MSE fell below a threshold of 0.001 or until a maximum of 1,000 epochs was reached, establishing a clear convergence criterion.

2. Adaptive Neuro-Fuzzy Inference System (ANFIS)

• Training Process:

• Model Setup: ANFIS integrates fuzzy logic with neural networks, utilizing a Sugeno-type rule-based system. The initial setup entails selecting input parameters and defining relevant membership functions.

• Clustering: A subtractive clustering method was employed to ascertain the number of fuzzy rules and corresponding membership functions. Each input data point served as a potential cluster center, contributing to the optimization of model complexity.

• Hyperparameter Tuning:

• Radius of Influence (RADII): The RADII parameter is integral to the subtractive clustering approach as it regulates the granularity of fuzzy partitioning. Through experimentation, optimal RADII values of 0.32 and 0.26 were identified for normal and stressed conditions, respectively.

• Training Duration: Consistent with the ANN approach, training continued until the MSE fell below the threshold of 0.001, or until a maximum of 1,000 epochs was reached, providing uniformity in stopping criteria.

3. Gene Expression Programming (GEP)

• Training Process:

• Genetic Algorithm Structure: GEP merges principles from genetic algorithms and genetic programming, initiating with a population of randomly generated chromosomes that represent potential solutions.

• Fitness Evaluation: The fitness of each chromosome is assessed based on its accuracy in predicting sunflower grain yield, typically measured using metrics such as MSE or RMSE.

• Hyperparameter Tuning:

• Population Size: The population size was fixed at 30 chromosomes, influencing genetic diversity and the potential for uncovering optimal solutions.

• Head Length and Gene Count: Default configurations were set with a head length of \(h =7\) and three genes per chromosome, based on the parameters established within the GeneXpro tools applied for GEP.

• Genetic Operators: Throughout various generations, a range of genetic operations—including mutation, crossover, and selection—were implemented. Specific operator settings were closely monitored to ensure a balanced trade-off between exploration (diversity) and exploitation (convergence toward optimal solutions).

Performance Evaluation and Final Model Selection

Cross-validation: Post-training, models were evaluated against the testing dataset to ascertain their generalizability to unseen data. Metrics such as (R2), RMSE, and MAE were utilized to gauge the accuracy of predictions.

Model Comparison: The derived performance indicators facilitated a comparative analysis among the ANN, ANFIS, and GEP models, allowing for the selection of the best-performing model based on criteria of accuracy and interpretability.

In summary, each model underwent meticulous training with a focus on hyperparameter tuning through diverse methodologies: trial-and-error for the ANN, parameter adjustments through clustering in ANFIS, and genetic operations in GEP. The overarching objective was to enhance model performance while ensuring robustness and generalizability in predicting sunflower grain yield across varying environmental conditions.

Answer to question five

5. Validation Methods: What methods were used to validate the models? Were cross-validation techniques employed?

Validation of ANN, ANFIS, and GEP Models:

The models (ANN, ANFIS, and GEP) were validated using a split-sample approach to assess predictive accuracy, generalization, and robustness. Below are the validation details:

1. Dataset Division:

• Training Set: 75% (216 samples) used for model training.

• Testing Set: 25% (72 samples) reserved for independent evaluation to ensure generalization and avoid data leakage during training.

• This division effectively mirrored real-world conditions, where models must predict outcomes for unseen data.

2. Performance Metrics

The models were evaluated using three key metrics:

• Coefficient of Determination (R²): This indicates how well the model predictions align with observed values, with values closer to 1 reflecting better model performance.

• Root Mean Squared Error (RMSE): Measures the magnitude of prediction errors, penalizing larger errors more heavily.

• Mean Absolute Error (MAE): This provides a more balanced evaluation by averaging absolute prediction errors, treating both small and large errors equally.

3. Cross-Validation Techniques:

Although cross-validation (e.g., k-fold) was not explicitly used, the following steps ensured robust model evaluation:

• Train-Test Split: Mimics cross-validation by using a separate testing set to evaluate model generalizability.

• Repeated Trials: Hyperparameters were optimized through multiple iterations of training, ensuring model consistency across different configurations.

• Sensitivity Analysis (GEP Model): Identified key input features (e.g., leaf length, head diameter) that significantly influenced model predictions, indirectly validating the relevance of selected features.

4. Graphical Validation:

• Scatter Plots (Predicted vs. Observed): Visualized how closely model predictions matched actual yield values, with regression lines and R² values showing prediction accuracy.

• Taylor Diagrams: Combined correlation, RMSE, and standard deviation into a single graph for a comprehensive performance assessment.

• Violin Plots: Compared distributions of predicted and observed values, highlighting the models' ability to capture data variability.

5. Key Findings from Validation:

• GEP Model:

Best performance with R² = 0.803 (normal conditions) and R² = 0.743 (salinity stress), showing the highest accuracy, lowest RMSE, and MAE across conditions. Its explicit mathematical equations allowed for deeper interpretation and validation.

• ANFIS Model:

Achieved intermediate accuracy with R² = 0.758 (normal) and R² = 0.668 (salinity stress). This model handled nonlinear relationships well through fuzzy logic, but its performance was slightly lower than the GEP model.

• ANN Model:

Lower performance with R² = 0.716 (normal) and R² = 0.634 (salinity stress), possibly due to its black-box nature, which made it more prone to overfitting and less interpretable than the other models.

The GEP model outperformed both ANFIS and ANN models in terms of predictive accuracy and robustness under both normal and salinity stress conditions. The validation process using multiple metrics and graphical methods confirmed the reliability of the models, with GEP offering the most effective solution for sunflower yield prediction under varying environmental conditions.

Answer to six question

In this study, sensitivity analysis was not conducted, and the parameters (LL, HD, HSW, and HDW) were identified as influential characteristics affecting sunflower seed yield and modeling using the GEP model under salt stress conditions, rather than as components related to sensitivity analysis. The following sections will elaborate on the role and significance of these parameters in impacting sunflower performance under salinity stress.

Leaf length is crucial due to its direct relationship with the plant's potential photosynthetic capacity. Longer leaves are capable of absorbing more sunlight, which is vital for the process of photosynthesis (Ren et al., 2019; Niinemets, 2010). Under salinity stress conditions, plants may experience reduced water availability and increased osmotic pressure, leading to physiological responses that limit growth. Therefore, maintaining effective leaf length is essential for maximizing photosynthesis and the overall efficiency of the plant (Ahmad et al., 201

---

## [Decision Letter · Decision Letter 2]

31 Jan 2025

Advanced Computational Approaches for Predicting Sunflower Yield: Insights from ANN, ANFIS, and GEP in Normal and Salinity Stress Environments

PONE-D-24-00132R2

Dear Dr. Kordrostami,

We’re pleased to inform you that your manuscript has been judged scientifically suitable for publication and will be formally accepted for publication once it meets all outstanding technical requirements.

Kind regards,

Morteza Taki, Ph.D

Academic Editor

PLOS ONE

**Comments to the Author**

1. If the authors have adequately addressed your comments raised in a previous round of review and you feel that this manuscript is now acceptable for publication, you may indicate that here to bypass the “Comments to the Author” section, enter your conflict of interest statement in the “Confidential to Editor” section, and submit your "Accept" recommendation.

Reviewer #3: All comments have been addressed

Reviewer #5: All comments have been addressed

2. Is the manuscript technically sound, and do the data support the conclusions?

Reviewer #3: Yes

Reviewer #5: Yes

3. Has the statistical analysis been performed appropriately and rigorously? 

Reviewer #3: (No Response)

Reviewer #5: Yes

4. Have the authors made all data underlying the findings in their manuscript fully available?

Reviewer #3: Yes

Reviewer #5: Yes

5. Is the manuscript presented in an intelligible fashion and written in standard English?

Reviewer #3: Yes

Reviewer #5: Yes

6. Review Comments to the Author

Reviewer #3: Dear authors

I evaluated the paper. All the comments were addressed and I think the paper can be published

Reviewer #5: (No Response)

7. PLOS authors have the option to publish the peer review history of their article (what does this mean? ). If published, this will include your full peer review and any attached files.

**Do you want your identity to be public for this peer review?** For information about this choice, including consent withdrawal, please see our Privacy Policy .

Reviewer #3: No

Reviewer #5: No

---

## [Editor Report · Acceptance letter]

PONE-D-24-00132R2

PLOS ONE

Dear Dr. Kordrostami,

I'm pleased to inform you that your manuscript has been deemed suitable for publication in PLOS ONE. Congratulations! Your manuscript is now being handed over to our production team.

Kind regards,

on behalf of

Dr. Morteza Taki

Academic Editor

PLOS ONE